# A non-Abelian parton state for the $\nu = 2 + 3/8$ fractional quantum Hall effect

**Ajit Coimbatore Balram**$^\star$

Institute of Mathematical Sciences, HBNI, CIT Campus, Chennai 600113, India

$\star$ cb.ajit@gmail.com

## Abstract

Fascinating structures have arisen from the study of the fractional quantum Hall effect (FQHE) at the even denominator fraction of 5/2. We consider the FQHE at another even denominator fraction, namely $\nu = 2 + 3/8$, where a well-developed and quantized Hall plateau has been observed in experiments. We examine the non-Abelian state described by the "$\bar{3}\bar{2}^2 1^4$" parton wave function and numerically demonstrate it to be a feasible candidate for the ground state at $\nu = 2 + 3/8$. We make predictions for experimentally measurable properties of the $\bar{3}\bar{2}^2 1^4$ state that can reveal its underlying topological structure.

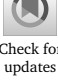

## Contents

Strongly interacting two-dimensional electron systems at low-temperatures and subjected to high perpendicular magnetic fields exhibit a broad range of non-perturbative phenomena. A case in point is the fractional quantum Hall effect (FQHE) [3,4] which arises from the Coulomb repulsion between electrons and leads to the formation of incompressible states at Landau level (LL) fillings $\nu$. The $\nu = 5/2$ FQHE, the first even denominator state to be observed [5], has produced a wide variety of remarkable concepts. Its most plausible explanation is in terms of the Moore-Read Pfaffian wave function [6] or its particle-hole conjugate, the anti-Pfaffian [7, 8]. The Pfaffian and anti-Pfaffian states represent topological $p$-wave paired states of fully spin-polarized composite fermions (CFs) [9], which are bound states of electrons and vortices [10]. Intriguingly, these states host non-Abelian excitations which could potentially be used to carry out fault-tolerant topological quantum computation [11].

This article is concerned with the physical origin of another even denominator fraction, namely $\nu = 2+3/8$, for which convincing experimental evidence exists [12–16]. In particular, Kumar *et al.* [15] demonstrated activated magnetotransport at this fraction, fully confirming the formation of an incompressible state here. Theoretically, the $2+3/8$ state was first studied in Ref. [17] where it was suggested that the inter-CF interaction could induce FQHE, however, the resulting paired state was shown to be distinct from the Pfaffian state. Subsequently, the anti-Pfaffian pairing of CFs was also ruled out at $2+3/8$ [18]. Hutasoit *et al.* [18] considered a Bonderson-Slingerland (BS) state and showed that the ground state $2 + 3/8$ is well-described by it. Like the Pfaffian and anti-Pfaffian states, the BS state also supports excitations that are expected to possess non-Abelian braid statistics [19]. In this work, we propose that the $2+3/8$ state could be described by a non-Abelian parton wave function which is topologically distinct from the BS state. We show that the parton state is in close competition with the BS state. Furthermore, to tell the two states apart we make predictions for many experimentally measurable properties. Definitive confirmation of the parton state at $2+3/8$ would lend further credence to the compelling assertion that *all* the experimentally observed FQHE states in the second LL (SLL) of GaAs conform to the parton paradigm [20].

The parton theory [21] (for a review of parton states see Sec. 1.2) has seen a resurgence in recent years owing to the following observations:

- *All* the FQHE states observed in the second Landau level of GaAs plausibly lend themselves to a description in terms of partons. The $\bar{n}\bar{2}1^3$ states (this notation is elucidated in detail in Sec. 1.2) for $n = 1, 2, 3$ capture the FQHE seen at $8/3$, $5/2$ and $2+6/13$ respectively [1,22,23] (see also Appendix B where we show new results to further demonstrate the viability of the $\bar{3}\bar{2}1^3$ state for $2 + 6/13$). Furthermore, the $\bar{n}\bar{2}^21^4$ states for $n = 1, 2$ capture the experimentally observed plateaus at $5/2$ and $12/5$ [20]. The $\bar{n}\bar{2}1^3$ and $\bar{n}\bar{2}^21^4$ sequences correspond to states at $\nu = 2n/(5n-2)$ and $n/(3n-1)$ respectively and FQHE in the SLL has been observed up to $n = 3$ at *all* these fillings. In GaAs quantum wells, FQHE has *only* been observed at the aforementioned fillings in the SLL aside from fractions that correspond to the $n/(4n \pm 1)$ sequence. The SLL states in the $n/(4n \pm 1)$ sequence are believed to be analogous to their LLL counterparts [24] (See Appendix A where we show results supporting this belief for $1/5$, $2/7$ and $2/9$.). The $\bar{n}\bar{2}1^3$ and $\bar{n}\bar{2}^21^4$ are related to each other by the symmetric state $\bar{2}1$, i.e., they are the $p = 1, 2$ members of the $\bar{n}1^2(\bar{2}1)^p$ family of states respectively. This is analogous to the primary and secondary Jain states, described respectively by the $n1^2$ and $n1^4$ states ($p = 1, 2$ members of the $n1^{2p}$ family of Jain states), which are related by the symmetric factor $1^2$. Unlike the factor $1^2$ which lends itself to a picture in terms of attachment of vortices/fluxes, we do not know of a simple interpretation for the factor $\bar{2}1$.

- The $n\bar{n}1^3$ states, which possess a novel $\mathbb{Z}_n$ topological order that violates the bulk-edge correspondence [6], could potentially be relevant to the $7/3$ FQHE [25, 26].

- The $\bar{2}^k 1^{k+1}$ states lie in the same universality class as the particle-hole conjugate of the $k$-cluster Read-Rezayi [27] (anti-RR$k$ or aRR$k$) states [20]. A nice feature of these parton states is that their wave functions can be evaluated for very large system sizes of the order of $N = 100$. In contrast, the wave functions of the Read-Rezayi states can be evaluated only for systems sizes of the order of $N = 30$.

- The 221 and $221^3$ states may apply to certain even-denominator FQHE states observed in graphene [28, 29] and wide quantum wells [30] respectively.

- The $\bar{3}^2 1^3$ state at 3/7 was considered in Ref. [2] and was shown to be feasible at and in the vicinity of the SLL (see also Appendix C where we show new results that further support the viability of the $\bar{3}^2 1^3$ state in the SLL). Although FQHE has not been established at $\nu = 2 + 3/7$, some signatures of it have been reported [14].

In light of these exciting recent developments in the parton theory, we revisit the FQHE observed at $2 + 3/8$. We consider the $n = 3$ member of the $\bar{n}\bar{2}^2 1^4$ sequence and show that it is a viable candidate to capture the ground state at $2 + 3/8$. Our results exhibit that the $2 + 3/8$ FQHE dovetails nicely with the parton description of SLL states.

Experimentally, FQHE has been well-established at $\nu = 2 + 3/8$ [12–16] only in GaAs quantum wells. The LL corresponding to the $n = 1$ orbital in the zeroth LL of bilayer graphene (BLG) is expected to be similar to the SLL of GaAs. Indeed, almost all FQHE states seen in the SLL of GaAs or their particle-hole conjugates have also been observed in BLG [31]. To the best of our knowledge, the only exception happens to be 3/8, where FQHE has been observed in the SLL of GaAs but surprisingly no FQHE is seen either at 3/8 or its particle-hole conjugate filling 5/8 in BLG [31]. The absence of FQHE at these fractions in BLG likely results from the fact that the single-particle wave function for the $n = 1$ orbital in the zeroth LL of BLG has an admixture of the $n = 0$ LL of ordinary semiconductors. Thus the effective interaction between the electrons is modified from the pristine $n = 1$ LL one. Presumably, the modified interaction leads to the formation of a bubble crystal ground state at 3/8 and 5/8 in BLG. It is possible that with a small change in the interaction between the electrons (due to screening by gates and/or LL mixing, etc.) the FQHE liquid at 3/8 could emerge in BLG.

This article is organized as follows: In Sec. 1 we provide some background material on the spherical geometry, a primer on the parton states, and introduce the candidate parton wave function for the $2 + 3/8$ ground state. In Sec. 2 we present results obtained from variational Monte Carlo (Sec. 2.1) and exact diagonalization (Sec. 2.2) calculations of the 3/8 state in the SLL. In Sec. 3 we present the experimental ramifications of our work and conclude the paper in Sec. 4 with a summary of our results.

# 1 Background

## 1.1 Spherical geometry

Comparisons with exact states available for finite systems have played an important role in confirming or eliminating various candidate FQHE states. For this purpose, we will employ Haldane's spherical geometry [32] where $N$ electrons reside on the surface of a sphere and a magnetic monopole of strength $2Q\phi_0$ (where $\phi_0 = hc/e$ is a flux quantum) produces a radial magnetic field. The radius of the sphere $R = \sqrt{Q}\ell$, where $\ell = \sqrt{\hbar c/(eB)}$ is the magnetic length and $B$ is the perpendicular magnetic field. In the LL indexed by $n$, the total number of single-particle orbitals is $2l + 1 = 2(Q + n) + 1$. Quantum Hall ground states on the sphere are uniform, i.e., have total orbital angular momentum $L = 0$. An incompressible state at a

filling factor $\nu$ occurs at $2l = \nu^{-1}N - \mathcal{S}$, where $\mathcal{S}$ is a topological quantum number called the shift [33]. Often candidate states at the same filling factor occur at different shifts.

In this work, we shall evaluate the ground-state energies of different candidate states to determine which among them is energetically favored. The total energy includes the contribution of the positively charged background which we assume interacts via the $1/r$ Coulomb potential. Assuming a uniform distribution of the background charge on the sphere, the electron-background and background-background interactions collectively contribute $-N^2/(2\sqrt{Q})\ell$ to the energy. We multiply the per-particle energies by a factor of $\sqrt{2Q\nu/N}$ before extrapolating them to the thermodynamic limit [34]. This factor corrects for the deviation of the electron density of a finite system from its $N \to \infty$ value, thereby providing a more accurate extrapolation. All the energies are quoted in units of $e^2/(\epsilon\ell)$, where $\epsilon$ is the dielectric constant of the host.

An important feature of an incompressible FQHE state is the existence of a finite gap to neutral and charged excitations. The neutral gap is defined as the difference between the two lowest energy states at a given value of $N$ and $2l$. The charge (or transport) gap is defined as the energy required to create a far-separated pair of fundamental (smallest magnitude charge) quasiparticle and quasihole. From exact diagonalization, the charge gap for a system of $N$ electrons at a given value of $2l$ can be obtained as:

$$
\begin{aligned}
\Delta^{\text{charge}} &= \frac{\mathcal{E}(2l-1) + \mathcal{E}(2l+1) - 2\mathcal{E}(2l)}{n_q}, \\
\mathcal{E}(2l) &= E(2l) - N^2 \frac{\mathcal{C}(2l)}{2}.
\end{aligned}
\tag{1}
$$

Here $E(2l)$ is the exact ground state energy of $N$ electrons at $2l$, $\mathcal{C}(2l)$ is the average charging energy at $2l$ which accounts for the background contribution [23], and $n_q$ is the number of fundamental quasiholes (quasiparticles) produced upon the insertion (removal) of a single flux quantum in the ground state. As mentioned above, for the $1/r$ Coulomb interaction, $\mathcal{C}(2l) = 1/\sqrt{l-n}\ e^2/(\epsilon\ell) = 1/\sqrt{Q}\ e^2/(\epsilon\ell)$.

Next, we state the approximations involved in our calculations which are routinely deployed in numerical studies of FQHE systems. In experiments, the finite magnetic field leads to LL mixing which breaks the degeneracy between a state and its particle-hole conjugate. However, throughout this work, we will make the simplifying assumption of neglecting LL mixing and thereby treat states related by particle-hole conjugation at the same footing. In the absence of LL mixing, we can focus our attention on a single LL. We shall discuss below all physics within the LLL subspace, even though we are interested in the second LL physics. This is possible because the problem of electrons in the SLL interacting via the Coulomb interaction is mathematically equivalent to the problem of electrons in the LLL interacting with an effective interaction that has the same Haldane pseudopotentials [32] as the Coulomb interaction in the second LL. In this work, we use the effective interaction given in Ref. [35] to simulate the physics of the SLL in the LLL. Aside from the spherical pseudopotentials, we shall also show results obtained from the disk pseudopotentials which are believed to provide a more reliable approach to the thermodynamic limit. Unless otherwise stated we shall assume the electrons to be fully spin-polarized. We will also neglect the effects of screening and disorder, which alter the form of the interaction and produce corrections to various observable quantities. Studies that take into account these effects will be needed for a more detailed and quantitative comparison with experiments.

## 1.2 Parton states

The parton theory, introduced by Jain [21], provides a scheme to construct candidate FQHE states. In the parton approach, one envisages dividing each electron into fictitious sub-particles

called "partons," placing each species of partons in an integer quantum Hall effect (IQHE) state with filling $n_\beta$ (here $\beta$ labels the different parton species), and finally sticking the partons back together to recover the physical electrons. The resulting parton state is denoted by "$n_1 n_2 \cdots$" and its wave function is given by

$$\Psi_\nu^{n_1 n_2 \cdots} = \mathcal{P}_{\text{LLL}} \prod_\beta \Phi_{n_\beta}(\{z_k\}). \tag{2}$$

Here $\Phi_n$ is the Slater determinant wave function of the IQHE state with $n$ filled LLs of electrons, $z_k = x_k - i y_k$, $k = 1, 2, \cdots, N$ is the two-dimensional coordinate of the $k^{\text{th}}$ electron parametrized as a complex number, and $\mathcal{P}_{\text{LLL}}$ projects the state into the LLL. We will denote a negative integer as $\bar{n} = -n$ with $\Phi_{\bar{n}} \equiv \Phi_{-n} = [\Phi_n]^*$. Note that each of the constituent IQHE states is itself made up of *all* of the electrons. The charge of the $\beta$ parton species is given by $e_\beta = -\nu e / n_\beta$, which is consistent with the constraint that charges of the partons add to that of the electron, i.e., $\sum_\beta e_\beta = -e$, where $-e$ is the charge of the electron. The wave function given in Eq. (2) occurs at the filling factor $\nu = \left[\sum_\beta n_\beta^{-1}\right]^{-1}$ and has a shift [33] $\mathcal{S} = \sum_\beta n_\beta$ in the spherical geometry. Thus the shift of any parton state is always an integer and therefore FQHE states with a non-integral shift [36–38] cannot be directly (not allowing for operations like particle-hole conjugation) obtained from a parton construction. One can generalize the parton construction to allow the partons themselves to form FQHE states (which can have a fractional shift [37, 38]) that can then result in FQHE states of electrons with a non-integral shift.

Many well-known FQHE states such as the Laughlin and Jain (composite fermion) states can be obtained from the parton construction. The $\nu = 1/(2p+1)$ Laughlin state is a $(2p+1)$-parton state where each of the partons forms a $\nu = 1$ IQHE state. The Laughlin state is denoted as "$11 \cdots 1$" [$(2p+1)$ 1s] and its wave function is given by $\Psi_{1/(2p+1)}^{\text{Laughlin}} = \Phi_1^{2p+1}$. The $\nu = n/(2pn \pm 1)$ Jain state is a $(2p+1)$-parton state where $2p$ partons form a $\nu = 1$ IQHE state and a single parton forms a $\nu = \pm n$ IQHE state. The Jain state is denoted as "$\pm n 11 \cdots 1$" [$2p$ 1s] and its wave function is given by $\Psi_{n/(2pn \pm 1)}^{\text{Jain}} = \mathcal{P}_{\text{LLL}} \Phi_{\pm n} \Phi_1^{2p}$. The parton theory allows us to construct states which go beyond the CF description. As we shall see below, the parton state of our interest is of the non-CF kind.

## 1.3 Trial states at 3/8

In this article, we shall concern ourselves with the $n = 3$ member of the $\bar{n}\bar{2}^2 1^4$ sequence of states at $\nu = n/(3n-1)$ which was posited in Ref. [20]. The $n = 1$ and $n = 2$ members of this sequence lie in the same phase as the anti-Pfaffian [7, 8] at $\nu = 1/2$ [22] and the aRR3 state [27] at $\nu = 2/5$ [20], which respectively, likely describe the experimentally observed FQHE at $\nu = 5/2$ and $12/5$. The wave function of the $\bar{3}\bar{2}^2 1^4$ state, which occurs at $\nu = 3/8$, is given by:

$$\Psi_{3/8}^{\bar{3}\bar{2}^2 1^4} = \mathcal{P}_{\text{LLL}} [\Phi_3^*][\Phi_2^*]^2 \Phi_1^4 \sim \frac{\Psi_{3/5}^{\text{Jain}} [\Psi_{2/3}^{\text{Jain}}]^2}{\Phi_1^2}, \tag{3}$$

where the $\sim$ sign indicates that the states either side of the sign differ in the details of how the projection to the LLL is implemented. Although, the two wave functions either side of the $\sim$ sign differ microscopically, we expect that they describe the same topological phase [39]. A nice feature of the Jain wave functions is that they can be evaluated for hundreds of electrons in *real space* (in first quantized form) using the Jain-Kamila projection [40–44]. Therefore, the form of the $\bar{3}\bar{2}^2 1^4$ wave function stated on the right-most end of Eq. (3) allows accessibility to large system sizes and would be used throughout this work. The shift of the above state in the spherical geometry is $\mathcal{S}^{\bar{3}\bar{2}^2 1^4} = -3$. The $n = 4$ member of the $\bar{n}\bar{2}^2 1^4$ sequence produces

a state at $\nu = 4/11$ where FQHE has not yet been observed in the SLL. We note that there is clear evidence for FQHE at $\nu = 4/11$ in the LLL [45, 46] and for its description, a parton state was recently proposed [47].

Another candidate state for the $2+3/8$ FQHE is the Bonderson-Slingerland (BS) state [19] that is described by the wave function:

$$\Psi^{BS}_{3/8} = \mathcal{P}_{LLL} Pf\left(\frac{1}{z_i - z_j}\right)[\Phi_3^*]\Phi_1^3 \sim Pf\left(\frac{1}{z_i - z_j}\right)\Phi_1 \Psi^{Jain}_{3/5}, \tag{4}$$

where Pf is the Pfaffian of an anti-symmetric matrix with the square of the Pfaffian being the determinant. The shift of the BS state, $\mathcal{S}^{BS} = 1$, is different from that of the state given in Eq. (3) indicating that the two states carry different topological orders [33]. The wave function of Eq. (4) was shown to be a good candidate to describe the $2 + 3/8$ FQHE [18]. In particular, for the only system of $N = 12$ electrons accessible to exact diagonalization, the 3/8 BS state has a good overlap of about 80% with the SLL Coulomb ground state [18]. The 3/8 BS state is the $n = 3$ member of the family of states defined by the BS wave function $\Psi^{BS}_{n/(3n-1)} = \mathcal{P}_{LLL} Pf\left([z_i - z_j]^{-1}\right)[\Phi_n^*]\Phi_1^3 \sim Pf\left([z_i - z_j]^{-1}\right)\Phi_1 \Psi^{Jain}_{n/(2n-1)}$, which describes states at $\nu = n/(3n-1)$ [same fillings as the $\bar{n}\bar{2}^2 1^4$ sequence mentioned above]. The $n = 1$ member is the same as the Moore-Read Pfaffian state at $\nu = 1/2$ [6]. The $n = 2$ member describes a state at $\nu = 2/5$ and has been put forth as a candidate state to describe the 12/5 FQHE [48]. The $n = 4$ member provides a state at $\nu = 4/11$ where FQHE has not yet been observed in the SLL.

## 2 Results

### 2.1 Variational Monte Carlo

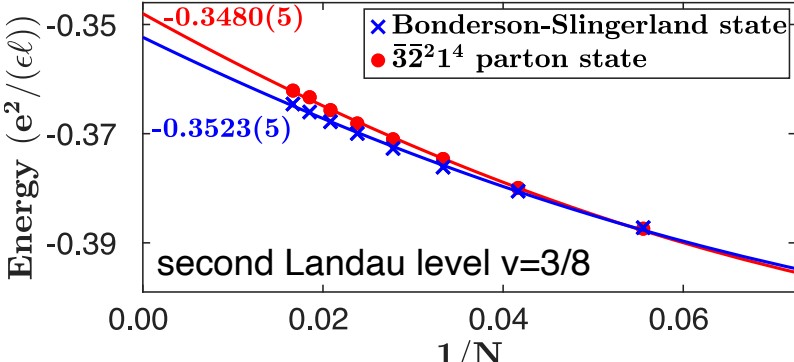

Figure 1: (color online) Thermodynamic extrapolations of the per-particle Coulomb energies for the Bonderson-Slingerland state (blue crosses) and the $\bar{3}\bar{2}^2 1^4$ state (red dots) for $\nu = 3/8$ in the second Landau level. The extrapolated energies, obtained from a quadratic fit in $1/N$, are quoted in Coulomb units of $e^2/(\epsilon\ell)$ on the plot. The energies include contributions of the background positive charge and have been density-corrected [34].

We compare the Coulomb energies of the $\bar{3}\bar{2}^2 1^4$ parton and the 3/8 BS states in the second Landau level in Fig. 1. We find that the two states are energetically competitive with each other with the BS state having slightly lower energy in the thermodynamic limit. We mention

here a couple of important caveats concerning these energetic comparisons. Firstly, in our variational calculations, we have made several approximations such as neglecting the effects of LL mixing, finite-width of the quantum well, screening, and disorder. When candidate states are close in energy, the precise nature of the ground state stabilized in experiments can only be determined by taking into account these effects. This is clearly beyond the scope of the current work. Secondly, due to technical difficulties, the projection of the parton and BS states was carried out as stated in the extreme right-hand side of Eqs. (3) and (4) respectively. Although we expect these versions of the wave functions to lie in the right topological phase, they may not be their best microscopic representatives. Therefore, even though the 3/8 BS state has slightly lower energy than the $\bar{3}\bar{2}^{2}1^{4}$ state for the exact second LL Coulomb point, it does not necessarily imply that the experimentally observed $2 + 3/8$ state is in the same topological phase as the BS state.

An analogous situation arises for the 12/5 FQHE where two different topological orders are in close competition with each other. The 2/5 BS state [19] and the aRR3 state [27] have nearly identical energies in the second Landau level and both states provide fairly good representations of the exact SLL Coulomb ground state [27, 48]. However, recent studies of the entanglement spectra of the 12/5 Coulomb ground state indicate that it likely lies in the same topological phase as the aRR3 state [49–51]. Recently, in Ref. [20] it was shown that the $\bar{2}^{3}1^{4}$ state lies in the same topological phase as the aRR3 state though the parton state does not provide as good a representation of the SLL Coulomb state as the aRR3 state does. Therefore, although the 2/5 BS state is lower in energy than the $\bar{2}^{3}1^{4}$ state in the SLL, the 12/5 Coulomb state likely lies in the same topological phase as that described by the $\bar{2}^{3}1^{4}$ state.

We conclude that our variational calculations are not able to decisively determine the nature of the ground state at $2 + 3/8$. This situation in the SLL should be contrasted with that in the LLL where variational energetic comparisons based on the CF theory can decisively determine the nature of the ground state. The reason is that unlike the trial states in the SLL, the CF states provide a near-perfect representation of the accessible LLL Coulomb ground states obtained from exact diagonalization [42, 52–54].

## 2.2 Exact diagonalization

Next, we present results obtained from exact diagonalization in the SLL. The shifts of the competitive candidate states can be identified from the existence of robust charge and neutral gaps and the presence of downward cusps in the ground-state energies for a fixed number of particles as the flux through the sphere is varied. In Fig. 2, we show the ground-state energy as well as the charge and neutral gaps for the smallest system of $N = 12$ electrons for $2l = 28$ to 39 in the SLL for both the spherical and disk pseudopotentials. We find a clearly discernible downward cusp in the ground-state energy at $2l = 35$ which corresponds to the proposed 3/8 parton state. Moreover, the state at $2l = 35$ harbors a robust charge and neutral gap. We find similar features at the values of $2l$ corresponding to other candidate states in this range, namely $2l = 33$ for 7/3 and $2l = 28$ for $2 + 6/13$. In contrast, at the value corresponding to the 3/8 BS state, $2l = 31$, we do not find a prominent downward cusp in the ground-state energy, or a robust charge gap. These results from exact diagonalization thus favor the parton description of $2 + 3/8$ over the BS state. The next system size for which the $\bar{3}\bar{2}^{2}1^{4}$ and 3/8 BS states can be constructed on the sphere is $N = 18$, which is currently not accessible to exact diagonalization since their Hilbert space dimension is over 324 billion and 60 billion respectively.

We now focus our attention on the system of $N = 12$ electrons at $2l = 35$ that has a Hilbert space dimension of about 16 million. For this system, we find that the exact Coulomb ground state in the SLL for both the spherical and disk pseudopotentials is uniform, i.e., has $L = 0$. The exact ground states obtained using the disk and spherical pseudopotentials have an overlap of

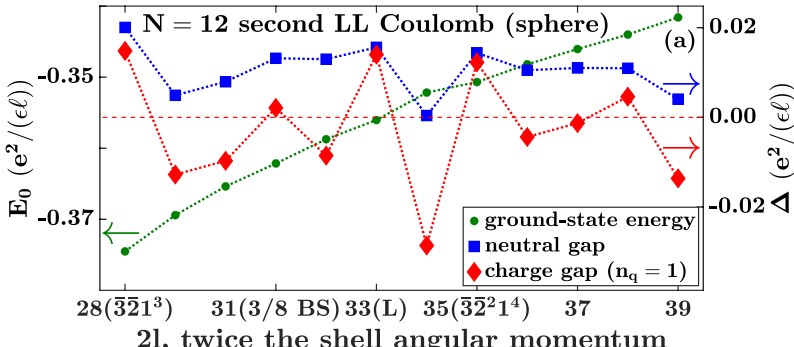

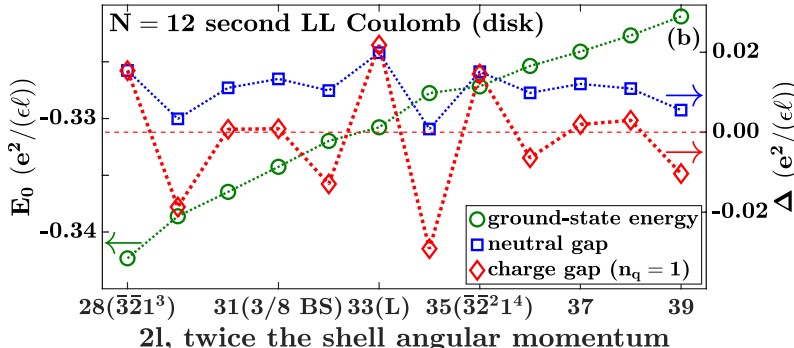

Figure 2: (color online) A plot of the second-Landau-level Coulomb ground-state energies (green dots) and neutral (blue squares) and charge (red diamonds) [using $n_q = 1$ in Eq. (1)] gaps for $N = 12$ electrons as a function of the shell-angular momentum $2l$ obtained using exact diagonalization in the spherical geometry with the spherical [top panel (a)] and disk [bottom panel (b)] pseudopotentials. A pronounced downward cusp in the ground-state energy and robust charge and neutral gaps are seen at $2l = 35$ which corresponds to the $\bar{3}\bar{2}^2 1^4$ state at 3/8. For reference we have also marked other states such as $2l = 28$ [$\bar{3}\bar{2}1^3$ state at $\nu = 6/13$ ($\bar{3}\bar{2}1^3$) [1]], $2l = 31$ [$\nu = 3/8$ Bonderson-Slingerland (3/8 BS) [19]] and $2l = 33$ [$\nu = 1/3$ Laughlin (L) [4]]. The dotted lines are a guide to the eye.

0.99 which indicates that they are quite close to each other. Due to technical reasons, it has not been possible for us to obtain the Fock-space representation (second quantized form) of the $\bar{3}\bar{2}^2 1^4$ state, which precludes an accurate calculation of its overlap with the exact ground states or a comparison of their entanglement spectrum [55] for even the $N = 12$ system [Using a Monte Carlo calculation we estimate that the overlap of $\bar{3}\bar{2}^2 1^4$ state [projected to the LLL as stated in Eq. (3)] with the exact Coulomb ground state obtained using the SLL disk pseudopotentials for $N = 12$ electrons to be 0.63(4). The number in the parenthesis indicates statistical uncertainty of the Monte Carlo estimate.]. However, we have compared the pair-correlation function $g(r)$ of the exact SLL Coulomb ground state with that of the $\bar{3}\bar{2}^2 1^4$ state for $N = 12$ electrons (see Fig. 3). The $g(r)$ of both states show oscillations that decay at long distances, which is a characteristic feature of an incompressible state [56–58]. The agreement between the $g(r)$ of the exact ground state and the parton state is on par with trial states at other filling factors in the SLL [23,59]. We note that the $\bar{3}\bar{2}^2 1^4$ state shows a "shoulder"-like feature in the $g(r)$ at short to intermediate lengths, which is considered a typical fingerprint of clustering in non-Abelian states [27]. For the $N = 12$ system, the exact energy for the effective

interaction we use to simulate the physics of the SLL in the LLL is $-0.4041$. In comparison, the $\bar{3}\bar{2}^2 1^4$ state has an energy of $-0.3993(2)$ for the same interaction, where the number in the parenthesis is the statistical uncertainty in the Monte Carlo estimate of the energy of the $\bar{3}\bar{2}^2 1^4$ state. Although not definitive, this level of agreement is comparable with that of other candidate states in the SLL [18, 23].

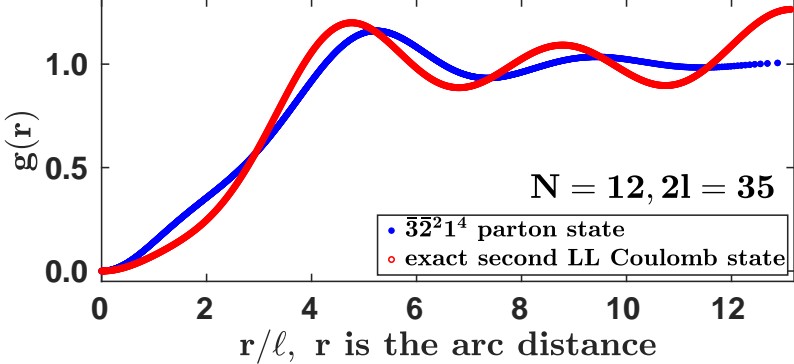

Figure 3: (color online) The pair correlation function $g(r)$ as a function of the arc distance $r$ on the sphere for the exact second Landau level Coulomb ground state (red filled dots), and the $\bar{3}\bar{2}^2 1^4$ state of Eq. (3) (blue open circles) for $N = 12$ electrons at $2l = 35$.

Next, we turn to charge and neutral gaps. Since only a single system size is accessible to exact diagonalization, we do not have estimates for the thermodynamic gaps of the $2 + 3/8$ state. Nevertheless, we have evaluated the gaps for the system of $N = 12$ electrons. The neutral gap is evaluated by taking the energy difference between the two lowest-energy states of $N = 12$ electrons at $2l = 35$. To calculate the charge gap, we use Eq. (1) with $n_q = 6$ since the insertion of a single flux quantum in the $\bar{3}\bar{2}^2 1^4$ state produces six fundamental quasiholes each of charge $e/16$. The neutral and charge gaps for $N = 12$, evaluated using exact diagonalization with both the spherical and disk pseudopotentials, are about 0.015 $e^2/(\epsilon \ell)$ and 0.003 $e^2/(\epsilon \ell)$ respectively. For comparison, these gaps are smaller than the corresponding gaps at $7/3$, which indicates that the $2+3/8$ state is more fragile compared to the $7/3$ state [23]. The gap calculations indicate strong finite-size effects in the second LL as evidenced by the fact that the neutral gap is larger than the charge gap. In the thermodynamic limit, we expect the charge gap to be greater than or equal to the neutral gap.

In the experiment of Ref. [15], the $2 + 3/8$ state is seen in a sample of width $w = 30$ nm at a magnetic field of $B = 5.2$ T which corresponds to $w/\ell = 2.7$. To incorporate the effect of the finite thickness of the quantum well, we consider a model in which the transverse wave function is taken to be the ground state for a particle in a box of width $w$ with hard-core walls. We calculate the pseudopotentials for this model of the finite-width interaction and carry out exact diagonalization with them [23]. Encouragingly, we find that the ground state of $N = 12$ at $2l = 35$ is uniform for both the spherical and disk pseudopotentials for (at least) $w \le 10\ell$. Moreover, the overlap between the exact ground states at $w = 0$ and $w = 10\ell$ is greater than 95% which suggests that the ground state is only weakly altered by finite-width corrections. Furthermore, as shown in Fig. 4 the ground state has a finite charge and neutral gap for all the widths considered with the gaps decreasing weakly with increasing widths. These results indicate that the ground state at $2 + 3/8$ is resistant to finite-width perturbations.

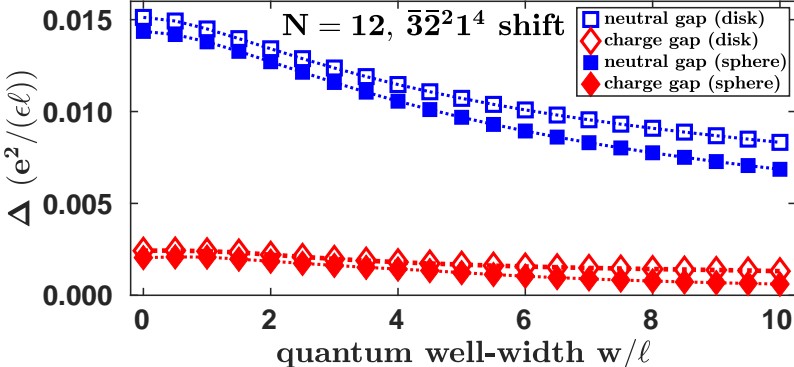

Figure 4: (color online) The second Landau level charge (red diamonds) and neutral (blue squares) gaps for $N = 12$ electrons at the shift corresponding to the $\bar{3}\bar{2}^2 1^4$ state at $\nu = 3/8$ evaluated in the spherical geometry using the spherical (filled symbols) and disk pseudopotentials (open symbols) for various well-widths $w$.

## 3  Experimental ramifications

We believe that the results shown in the previous section make a strong case for the plausibility of the $\bar{3}\bar{2}^2 1^4$ ansatz at $2+3/8$. In this section we deduce some of its experimental consequences which allow its validity to be assessed and provides ways to unambiguously distinguish it from the $3/8$ BS state. Owing to the repeated factor of $\bar{2}$, the quasiparticles of the $\bar{3}\bar{2}^2 1^4$ obey non-Abelian braid statistics [60]. An additional quasiparticle in the factor $\Phi_{\bar{3}}$ has charge $q_{\bar{3}} = -e/8$, whereas that in the factor $\Phi_{\bar{2}}$ has charge $q_{\bar{2}} = -3e/16$. A combination of a quasihole in $\Phi_{\bar{3}}$ and a quasiparticle in $\Phi_{\bar{2}}$ leads to the fundamental quasiparticle of the $\bar{3}\bar{2}^2 1^4$ state which carries a charge of $q_{\bar{2}} - q_{\bar{3}} = -e/16$. The $3/8$ BS state is also non-Abelian and its smallest charged quasiparticle also has a charge of $-e/16$ [18]. Due to the presence of the $\bar{3}$ factor, both the $\bar{3}\bar{2}^2 1^4$ and $3/8$ BS states support upstream neutral modes. Thus, a measurement of the charge of the fundamental quasiparticle or the presence of neutral modes does not allow us to distinguish the parton and BS states.

We now mention experimental measurements that can distinguish between the parton and BS orders. Owing to the different shifts, the parton and BS states have different Hall viscosities [61] $\eta_{\text{H}} = 3\hbar \mathcal{S}/(64\pi\ell^2)$, where $\mathcal{S}$ is the shift of the state in the spherical geometry [33] with $\mathcal{S}^{\bar{3}\bar{2}^2 1^4} = -3$ and $\mathcal{S}^{3/8 \text{ BS}} = 1$. Furthermore, *assuming* full equilibration of the edge modes the thermal Hall conductance of the $3/8$ BS state is $\kappa_{xy}^{\text{BS}} = -1/2[\pi^2 k_{\text{B}}^2/(3h)]T$ [18], which is different from that of the parton state, which has $\kappa_{xy}^{\bar{3}\bar{2}^2 1^4} = -5/2[\pi^2 k_{\text{B}}^2/(3h)]T$ [20] [the two lowest filled LLs with spin up and spin down provide an additional contribution of $2[\pi^2 k_{\text{B}}^2/(3h)]T$ to $\kappa_{xy}$]. The thermal Hall effect has been measured at certain fillings in the second Landau level [62] and thus could potentially distinguish the parton and BS states.

We also mention here other candidates that have been put forth for $3/8$. Jolicoeur [63] has proposed the following state at $3/8$:

$$\Psi_{3/8}^{\text{Jolicoeur}} = \mathcal{P}_{\text{LLL}}[\Psi_3^{\text{bosonic-RR}}]^* \Phi_1^3, \tag{5}$$

where the bosonic version of the six-cluster Read-Rezayi (RR) state [27] is defined as:

$$\Psi_3^{\text{bosonic-RR}} = \mathbb{S}[\prod_{i_1 < j_1} (z_{i_1} - z_{j_1})^2 \cdots \prod_{i_6 < j_6} (z_{i_6} - z_{j_6})^2], \tag{6}$$

where $\mathbb{S}$ denotes the operation of symmetrization of the $N$ particles over the six clusters. The Jolicoeur state has a shift of $\mathcal{S}^{\text{Jolicoeur}} = 1$, which is different from the $\bar{3}\bar{2}^2 1^4$ state but is

identical to that of the 3/8 BS state. The 3/8 Jolicoeur wave function is not easily amenable to a numerical calculation and its properties have not been studied in detail in the literature. Using the effective edge theory-based classification, Fröhlich *et al.* [64] obtained a chiral Abelian state at 3/8. However, there is no prescription to construct a trial wave function from this approach which precludes its comparison with numerics.

The anti-Pfaffian analog of the 3/8 BS state is given by:

$$\Psi_{3/8}^{\text{aPf}-\text{BS}} = \mathcal{P}_{\text{LLL}}\Psi_{1/2}^{\text{aPf}}[\Phi_3^*]\Phi_1, \tag{7}$$

where $\Psi_{1/2}^{\text{aPf}}$ is the anti-Pfaffian state at $\nu = 1/2$ [7, 8], which is the particle-hole conjugate of the Pfaffian state i.e., $\Psi_{1/2}^{\text{aPf}} = \mathcal{P}_{\text{ph}}\left(\text{Pf}\left[(z_i - z_j)^{-1}\right]\Phi_1^2\right)$, where $\mathcal{P}_{\text{ph}}$ denotes the operation of particle-hole conjugation. The anti-Pfaffian analog of the BS state has a shift of $\mathcal{S}^{\text{aPf}-\text{BS}} = -3$, which is the same as that of the state given in Eq. (3). Moreover, the thermal Hall conductance of the anti-Pfaffian analog of the 3/8 BS state is the same as that of the $\bar{3}\bar{2}^2 1^4$ state. Thus these states likely describe the same topological order. This can be understood by noting that the $\bar{2}^2 1^3$ state lies in the same universality class as the anti-Pfaffian [22]. Unlike the parton and BS states, we do not know of an efficient way to evaluate quantities for the anti-Pfaffian analog of the BS state since the square of the anti-Pfaffian, unlike the Pfaffian, cannot be written in a simple form.

Finally, we consider two-component states at 3/8, where the two components could represent spin, valley, layer, subband, or orbital degrees of freedom. Besides the fully polarized state, the $\bar{3}\bar{2}^2 1^4$ state admits the possibility of partially polarized and singlet states arising from the corresponding states at $\nu = -3$ and $\nu = -2$ respectively. On the other hand, the 3/8 BS state only has fully polarized and partially polarized states which stem from the corresponding states at $\nu = -3$. It is possible that for certain interaction parameters the non-fully polarized states become the ground state. Recently, an experiment using the technique of spin-resolved pulsed tunneling has indicated the presence of non-fully spin-polarized states in the second Landau level [65].

## 4 Conclusion

Many remarkable concepts such as Majorana modes obeying non-Abelian braid statistics have emerged from the study of the FQHE at 5/2. In this work, we looked at the only other even denominator filling factor in the SLL where experimentally an FQHE state has been well-established, namely $\nu = 2 + 3/8$. We considered the non-Abelian state described by the $\bar{3}\bar{2}^2 1^4$ wave function and showed it to be a viable candidate to capture the $2 + 3/8$ Coulomb ground state. Our analysis suggests that the $\bar{3}\bar{2}^2 1^4$ and the 3/8 Bonderson-Slingerland states are in close competition with each other at $2 + 3/8$. We also proposed experimental probes that can unambiguously distinguish the non-Abelian topological orders of the $\bar{3}\bar{2}^2 1^4$ and the 3/8 Bonderson-Slingerland states.

## Acknowledgments

We acknowledge useful discussions with Maissam Barkeshli, Jainendra K. Jain, Sutirtha Mukherjee, G. J. Sreejith, Arkadiusź Wójs, and Andrea Young. Computational portions of this research work were conducted using the Nandadevi supercomputer, which is maintained and supported by the Institute of Mathematical Science's High-Performance Computing Center. Some of the numerical calculations were performed using the DiagHam package, for which we are grateful to its authors.

**Funding information** We thank the Science and Engineering Research Board (SERB) of the Department of Science and Technology (DST) for funding support via the Start-up Grant SRG/2020/000154.

# A States of composite fermions carrying four vortices in the second Landau level

The most prominent FQHE states belonging to the $^4$CF sequence, $\nu = n/(4n \pm 1)$, that have been observed in the SLL are at filling factors $\nu = 2 + 1/5$ and $2 + 2/7$ and their particle-hole conjugates at $\nu = 2 + 4/5$ and $\nu = 2 + 5/7$ [13–16, 66]. In this Appendix, we present evidence to show that the 1/5 and 2/7 states are well-described by the 1/5 Laughlin and 2/7 Jain state respectively. This implies that the SLL states at $n/(4n\pm1)$ and their particle-hole conjugates are analogous to their LLL counterparts. Since the Hilbert space of systems at these low-fillings is quite large, it is computationally expensive to study these systems for a wide variety of interactions using exact diagonalization. Thus, we shall focus our attention on the exact SLL Coulomb point and present only results obtained using the spherical pseudopotentials.

In Table 1 we present the overlaps of the 1/5 Laughlin state with the exact Coulomb ground state at 1/5 in the two lowest Landau levels. The exact Coulomb ground states at 1/5 in the SLL has an overlap upwards of 0.92 with the Laughlin state for up to $N = 11$ (see also results of Refs. [24, 67]). We find that the overlaps of the Laughlin state in the SLL are comparable to the analogous numbers in the LLL. Moreover, the overlap between the LLL and SLL Coulomb ground state at 1/5 is almost unity for all the systems considered in this work.

Table 1: Absolute value of the overlap of the exact Coulomb ground state at $\nu = 1/5$ in the lowest Landau level (LLL), $|\Psi_{1/5}^{\text{LLL}}\rangle$ and second Landau level (SLL), $|\Psi_{1/5}^{\text{SLL}}\rangle$ with the 1/5 Laughlin state $|\Psi_{1/5}^{\text{Laughlin}}\rangle$ obtained in the spherical geometry for $N$ electrons at $2l = 5N - 5$. For comparison, in the last column, we have shown the overlap between the exact LLL and SLL states. The numbers in the third and fourth columns for up to $N = 10$ were previously given in Refs. [24, 67].

| $N$ | $2l$ | $|\langle\Psi_{1/5}^{\text{Laughlin}}|\Psi_{1/5}^{\text{LLL}}\rangle|$ | $|\langle\Psi_{1/5}^{\text{Laughlin}}|\Psi_{1/5}^{\text{SLL}}\rangle|$ | $|\langle\Psi_{1/5}^{\text{SLL}}|\Psi_{1/5}^{\text{LLL}}\rangle|$ |
|---|---|---|---|---|
| 6 | 25 | 0.9486 | 0.9590 | 0.9993 |
| 7 | 30 | 0.9768 | 0.9818 | 0.9996 |
| 8 | 35 | 0.9589 | 0.9678 | 0.9992 |
| 9 | 40 | 0.9334 | 0.9453 | 0.9992 |
| 10 | 45 | 0.9228 | 0.9386 | 0.9987 |
| 11 | 50 | 0.9413 | 0.9509 | 0.9993 |

In Table 2 we present the overlaps of the 2/7 Jain state with the exact Coulomb ground state at 2/7 in the two lowest Landau levels. The 2/7 Jain state was obtained by a brute-force projection of the unprojected state into the LLL. The exact Coulomb ground states at 2/7 in the LLL and SLL has an overlap of 89% or higher with the 2/7 Jain state for up to $N = 10$. Furthermore, the overlap between the LLL and SLL Coulomb ground state at 2/7 is also 91% or higher for all the systems considered in this work.

Besides 1/5 and 2/7, and their particle-hole conjugates there are no FQHE states in the sequence $n/(4n \pm 1)$ that have been definitively established in the SLL. Some signatures of FQHE have been observed at $2 + 7/9$, which is the particle-hole conjugate of the $2 + 2/9$ state [13, 15]. Therefore, for completeness, in Table 3 we present the overlaps of the 2/9 Jain

Table 2: Absolute value of the overlap of the exact Coulomb ground state at $\nu = 2/7$ in the lowest Landau level (LLL), $|\Psi_{2/7}^{\text{LLL}}\rangle$ and second Landau level (SLL), $|\Psi_{2/7}^{\text{SLL}}\rangle$ with the 2/7 Jain state $|\Psi_{2/7}^{\text{Jain}}\rangle$ obtained in the spherical geometry for $N$ electrons at $2l = 7N/2 - 2$. For comparison, in the last column, we have shown the overlap between the exact LLL and SLL states. We have not been able to construct the 2/7 Jain state for $N = 12$ electrons, so its overlap with the exact states is currently unavailable (indicated by $-$).

| $N$ | $2l$ | $|\langle\Psi_{2/7}^{\text{Jain}}|\Psi_{2/7}^{\text{LLL}}\rangle|$ | $|\langle\Psi_{2/7}^{\text{Jain}}|\Psi_{2/7}^{\text{SLL}}\rangle|$ | $|\langle\Psi_{2/7}^{\text{SLL}}|\Psi_{2/7}^{\text{LLL}}\rangle|$ |
|---|---|---|---|---|
| 4 | 12 | 0.9999 | 0.9992 | 0.9996 |
| 6 | 19 | 0.9964 | 0.9681 | 0.9833 |
| 8 | 26 | 0.9989 | 0.9762 | 0.9819 |
| 10 | 33 | 0.9882 | 0.8969 | 0.9403 |
| 12 | 40 | $-$ | $-$ | 0.9119 |

Table 3: Absolute value of the overlap of the exact Coulomb ground state at $\nu = 2/9$ in the lowest Landau level (LLL), $|\Psi_{2/7}^{\text{LLL}}\rangle$ and second Landau level (SLL), $|\Psi_{2/9}^{\text{SLL}}\rangle$ with the 2/9 Jain state $|\Psi_{2/9}^{\text{Jain}}\rangle$ obtained in the spherical geometry for $N$ electrons at $2l = 9N/2 - 6$. For comparison, in the last column, we have shown the overlap between the exact LLL and SLL states. We have not been able to construct the 2/9 Jain state for $N = 12$ electrons, so its overlap with the exact states is currently unavailable (indicated by $-$).

| $N$ | $2l$ | $|\langle\Psi_{2/9}^{\text{Jain}}|\Psi_{2/9}^{\text{LLL}}\rangle|$ | $|\langle\Psi_{2/9}^{\text{Jain}}|\Psi_{2/9}^{\text{SLL}}\rangle|$ | $|\langle\Psi_{2/9}^{\text{SLL}}|\Psi_{2/9}^{\text{LLL}}\rangle|$ |
|---|---|---|---|---|
| 4 | 12 | 0.9999 | 0.9992 | 0.9996 |
| 6 | 21 | 0.9928 | 0.9892 | 0.9991 |
| 8 | 30 | 0.9955 | 0.9928 | 0.9989 |
| 10 | 39 | 0.9744 | 0.9766 | 0.9977 |
| 12 | 48 | $-$ | $-$ | 0.9972 |

state with the exact SLL Coulomb ground state at 2/9. As with the 2/7 Jain state, we obtain the 2/9 Jain state by a brute-force projection of the unprojected state into the LLL. The exact Coulomb ground states at 2/9 in the LLL and SLL have an overlap of about 97% or higher with the 2/9 Jain state for up to $N = 10$. Moreover, the exact LLL and SLL Coulomb ground state at 2/9 are almost identical to each other for all the systems considered in this work. All these results strongly suggest that the SLL states at $n/(4n \pm 1)$ and their particle-hole conjugates are analogous to their LLL counterparts.

We now turn to the charge and neutral gaps obtained from exact diagonalization at $\nu = 1/5$, 2/7, and 2/9 in the SLL. In Fig. 5 we show the charge and neutral gaps at these three fillings in the two lowest LLs. For all the systems considered in this work, all three fillings support a finite charge and neutral gap in the two lowest LLs. However, since only a few systems are available, the estimated extrapolated gaps in many cases have large uncertainty. Interestingly, we find that the gaps at 1/5 in the SLL are larger than the corresponding gaps in the LLL. This should be contrasted with the 1/3 filling where the gap in the LLL is larger than that in the SLL [23]. We note that the order of magnitude of the gaps at $\nu = 1/5$, 2/7 and 2/9 in the LLL and SLL are comparable to each other.

To summarize, our results suggest that the nature of the states belong to the $^4$CF sequence in the two lowest Landau levels are similar. In particular, the topological properties of the $n/(4n \pm 1)$ states in the LLL and SLL are expected to be identical to each other.

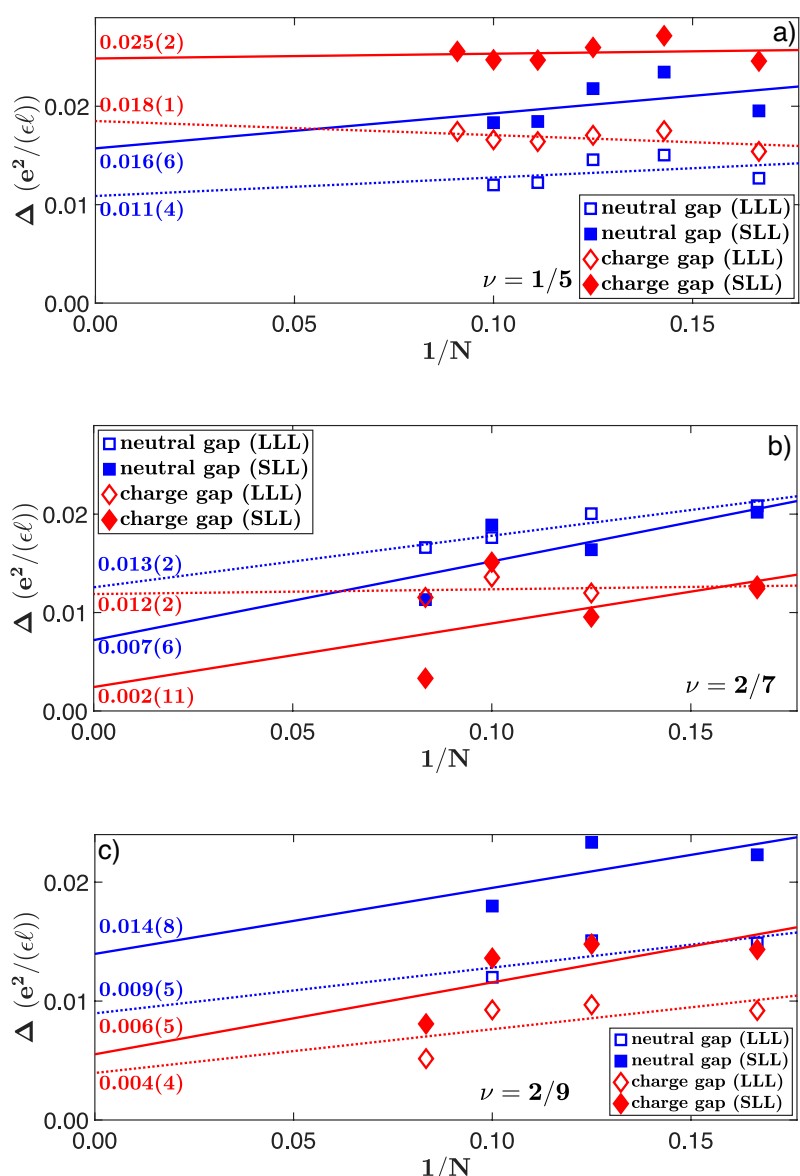

Figure 5: (color online) Thermodynamic extrapolations of the charge (red diamonds) and neutral (blue squares) gaps in the second Landau level (filled symbols) at $\nu = 1/5$ (left panel), 2/7 (center panel) and 2/9 (right panel) obtained from exact diagonalization in the spherical geometry. The extrapolated gaps, obtained from a linear fit in $1/N$, are quoted in Coulomb units of $e^2/(\epsilon\ell)$ on the plot, with the error in the extrapolation shown in the parenthesis. For comparison we have also shown the corresponding lowest Landau level gaps (open symbols). The lowest Landau level charge gaps at 1/5 and 2/9 were previously given in Ref. [68].

## B  Fractional quantum Hall effect at $\nu = 2 + 6/13$: an update on the results of Ref. [1]

The $\bar{3}\bar{2}1^3$ state has been proposed as a candidate [1] to describe the experimentally observed FQHE at $\nu = 2 + 6/13$ [15]. In Ref. [1], the $\bar{3}\bar{2}1^3$ state was constructed in real space for the smallest system of $N = 12$ electrons and was compared against the ground states obtained

from exact diagonalization of the SLL Coulomb as well as certain model interactions using the Monte Carlo method. These comparisons with exact states that were carried out using the real space representation of the parton state are time-consuming and computationally expensive. The Fock space representation readily allows a calculation of a state's overlap with ground states obtained from the exact diagonalization of various interactions. At the time of publication, it was not possible to obtain the Fock space representation of this parton state. We have now been able to obtain the Fock space representation of the $\bar{3}\bar{2}1^3$ state evaluated as $\Psi_{2/3}^{\text{Jain}}\Psi_{3/5}^{\text{Jain}}/\Phi_1$ for $N = 12$ electrons at $2l = 28$. We shall present some results obtained from it in this Appendix. We note that results obtained from exact diagonalization for the next system size of $N = 18$ electrons were shown in the supplemental material of Ref. [23]. However, due to the prohibitively large Hilbert space dimension, it has not been possible to obtain the Fock space representation of the $\bar{3}\bar{2}1^3$ state for $N = 18$.

To evaluate the Fock space representation, i.e., expansion coefficients in the $L_z = 0$ basis, of the desired state (which can be evaluated in real space), we follow the method outlined in Refs. [22, 69]. Since our desired state is uniform, we first calculate all the $L = 0$ states of the system of interest. To obtain all the $L = 0$ states, we evaluate a sufficient number of $L = 0$ states by starting with random initial vectors in the $L_z = 0$ basis and Lanczos diagonalizing the $L^2$ operator. We then Gram-Schmidt orthogonalize the states obtained in the previous step to get a complete set of orthonormal $L = 0$ states. Once the set of $L = 0$ states is obtained, we evaluate them as well as the desired state at sufficiently many (a few times the dimension of the $L = 0$ subspace) configurations $\{z_k\}$ to obtain a set of linear equations, which we then solve by the least-squares method (using the procedure of iterative refinement [70] implemented in the ALGLIB package [71]) to obtain the expansion coefficients. Depending on the chosen sets of $\{z_k\}$, the solution to the linear equations obtained can be numerically unstable, which is why we solve an over-determined system of equations. Note that each configuration $\{z_k\}$ gives two equations, one for the real part and one for the imaginary part since the expansion coefficients are chosen to be real. Instead of choosing the configurations completely at random, we find it useful to run a Monte Carlo with the desired state and picking configurations only after the Monte Carlo has thermalized.

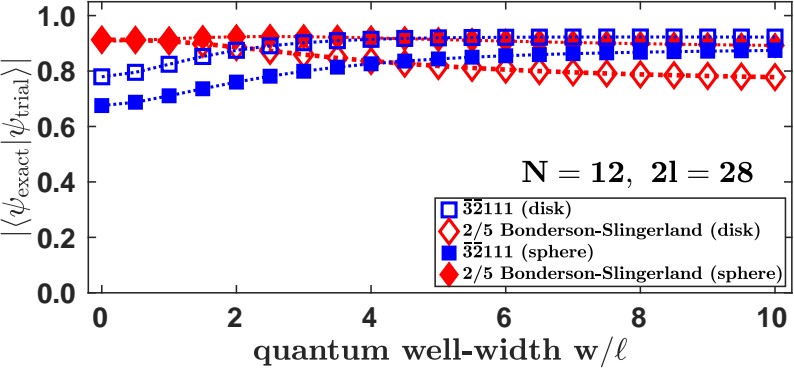

Figure 6: (color online) The overlap of the $\bar{3}\bar{2}1^3$ state (blue squares) with the exact second Landau level Coulomb ground state evaluated in the spherical geometry using the disk (open symbols) and spherical (filled symbols) pseudopotentials for various widths $w$ for $N = 12$ electrons at $2l = 28$. This system aliases with the 2/5 Bonderson-Slingerland state; therefore, for comparison, we have also shown its overlaps (red diamonds) with the exact second Landau level Coulomb ground states.

In Fig. 6 we show the overlap of the exact second LL Coulomb ground states obtained using the disk and spherical pseudopotentials with the $\bar{3}\bar{2}1^3$ state for a system of $N = 12$ electrons

at $2l = 28$ for $w \leq 10\ell$. We find that the $\bar{3}\bar{2}1^3$ state has a reasonable overlap with the exact SLL Coulomb ground state for all the widths considered. Furthermore, as the well-width $w$ is increased, the overlap of the $\bar{3}\bar{2}1^3$ state with the exact SLL Coulomb ground state increases which indicates that increasing well-width enhances the stability of the $\bar{3}\bar{2}1^3$ state. These results are consistent with previous results of Ref. [1], where the effect of the finite well-width was modeled by the Zhang-DasSarma interaction [72]. The system of $N = 12$ electrons at $2l = 28$ aliases with the 2/5 Bonderson-Slingerland state (BS) [19] which has been put forth as a candidate for the experimentally observed 12/5 FQHE [48]. Therefore, for completeness, in Fig. 6 we have also shown the overlap of the 2/5 BS state with the exact second LL Coulomb ground states. Consistent with previous results of Ref. [48], we find that the 2/5 BS state has a good overlap with the exact SLL Coulomb ground state.

Next, we present results on the charge and neutral gaps of $2 + 6/13$. For the gap calculations, we have only been able to access the system of $N = 12$ electrons using exact diagonalization. The neutral gap is evaluated by taking the energy difference between the two lowest-energy states of $N = 12$ electrons at $2l = 28$. To calculate the charge gap, we use Eq. (1) with $n_q = 6$ since the insertion of a single flux quantum in the $\bar{3}\bar{2}1^3$ state produces six fundamental quasiholes each of charge $e/13$ [1]. The neutral and charge gaps for $N = 12$, evaluated using exact diagonalization with both the spherical and disk pseudopotentials, are about $0.02\ e^2/(\epsilon\ell)$ and $0.002\ e^2/(\epsilon\ell)$ respectively at width, $w = 0$ and decrease with increasing $w$ as shown in Fig. 7. These gaps are smaller than the corresponding gaps at 7/3, which suggests that the $2 + 6/13$ state is more fragile compared to the prominent 7/3 state [23]. As pointed above, the system of $N = 12$ at $2l = 28$ aliases with the 2/5 BS state. The neutral gap corresponding to the 2/5 BS state is identical to that shown in Fig. 7. However, the charge gap corresponding to the 2/5 BS state is larger by a factor of three compared to that shown in Fig. 7 since the insertion of a single flux quantum in the 2/5 BS state produces $n_q = 2$ fundamental quasiholes each of charge $e/5$ [48].

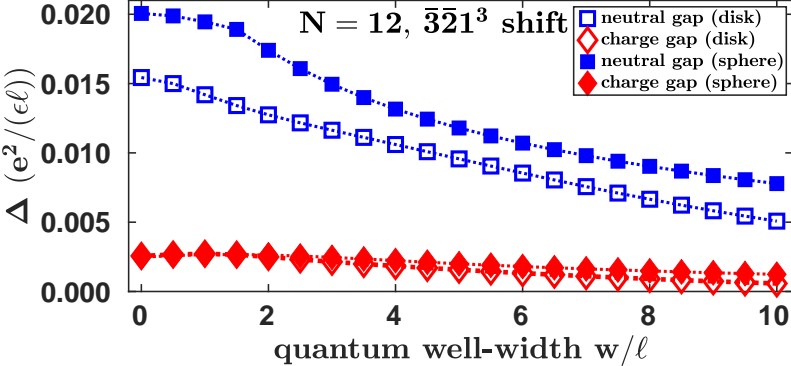

Figure 7: (color online) The second Landau level charge (red diamonds) and neutral (blue squares) gaps for $N = 12$ electrons at the shift corresponding to the $\bar{3}\bar{2}1^3$ state at $\nu = 6/13$ evaluated in the spherical geometry using the spherical (filled symbols) and disk pseudopotentials (open symbols) for various well-widths $w$.

In summary, our results suggest that the $\bar{3}\bar{2}1^3$ state gives a good description of the $2+6/13$ Coulomb ground state. Encouragingly, we find that the $\bar{3}\bar{2}1^3$ state is fairly robust to perturbations stemming from the finite-width of the quantum well.

# C  Fractional quantum Hall effect at $\nu = 2 + 3/7$: an update on the results of Ref. [2]

The $\bar{3}^2 1^3$ state has been proposed as a candidate [2] to describe an FQHE that could arise at $\nu = 2 + 3/7$. As yet, FQHE has not been definitely established at 3/7 in the second Landau level, though signatures of it have been seen in experiments [14]. In Ref. [2], the $\bar{3}^2 1^3$ state was constructed in Fock space for only the smallest system of $N = 9$ electrons. Using the method outlined in Appendix B, we have now been able to obtain the Fock space representation of the $\bar{3}^2 1^3$ state, evaluated as $[\Psi_{3/5}^{\text{Jain}}]^2/\Phi_1$, for the next system size of $N = 12$ electrons at $2l = 31$. We shall present some results obtained from it in this Appendix. We note that the state at $\nu = 3/7$ for sufficiently short-range interactions and the long-range Coulomb interaction in the LLL and the $n = 1$ LL of monolayer graphene, is the Abelian $\Psi_{3/7}^{\text{Jain}}$ CF state [42, 54, 73, 74].

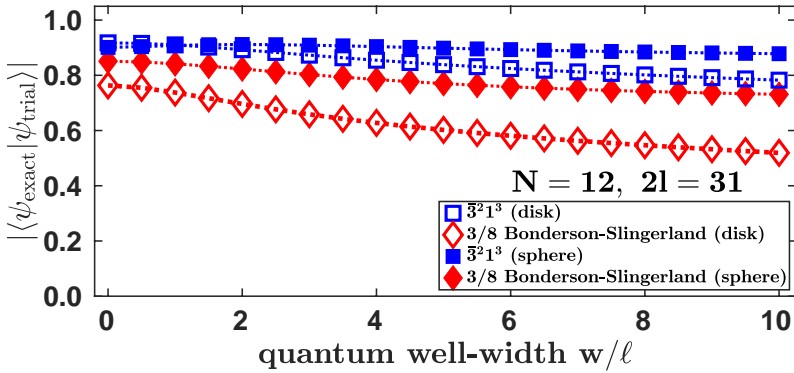

Figure 8: (color online) The overlap of the $\bar{3}^2 1^3$ state (blue squares) with the exact second Landau level Coulomb ground state evaluated in the spherical geometry using the disk (open symbols) and spherical (filled symbols) pseudopotentials for various widths $w$ for $N = 12$ electrons at $2l = 28$. This system aliases with the 3/8 Bonderson-Slingerland state; therefore, for comparison, we have also shown its overlaps (red diamonds) with the exact second Landau level Coulomb ground states.

In Fig. 8 we show the overlap of the exact second LL Coulomb ground states obtained using the disk and spherical pseudopotentials with the $\bar{3}^2 1^3$ state for a system of $N = 12$ electrons at $2l = 31$ for $w \le 10\ell$. We find that the $\bar{3}^2 1^3$ state has a reasonably high overlap with the exact SLL Coulomb ground state for all the widths considered. The system of $N = 12$ electrons at $2l = 31$ aliases with the 3/8 Bonderson-Slingerland state (BS) [19] which was discussed in detail in the main text. Therefore, for completeness, in Fig. 8 we have also shown the overlap of the 3/8 BS state with the exact second LL Coulomb ground states. Consistent with previous results of Ref. [18], that considered only the exact SLL Coulomb point in the spherical geometry, we find that the 3/8 BS state has a good overlap with the exact SLL Coulomb ground state.

Next, we present results on the charge and neutral gaps for the system of $N = 12$ electrons. The neutral gap is evaluated by taking the energy difference between the two lowest-energy states of $N = 12$ electrons at $2l = 31$. To calculate the charge gap, we use Eq. (1) with $n_q = 3$ since the insertion of a single flux quantum in the $\bar{3}^2 1^3$ state produces three fundamental quasi-holes each of charge $e/7$ [2]. The neutral and charge gaps for $N = 12$, evaluated using exact diagonalization with both the spherical and disk pseudopotentials, are about 0.015 $e^2/(\epsilon\ell)$ and 0.001 $e^2/(\epsilon\ell)$ respectively at width, $w = 0$ and decrease with increasing $w$ as shown in Fig. 9. These gaps are smaller than the corresponding gaps at experimentally observed frac-

tions in the SLL, which suggests that the $2 + 3/7$ state is quite fragile. As we mentioned in the previous paragraph, the system of $N = 12$ at $2l = 31$ aliases with the 3/8 BS state. The neutral gap corresponding to the 3/8 BS state is identical to that shown in Fig. 9. However, the charge gap corresponding to the 3/8 BS state is smaller by a factor of two compared to that shown in Fig. 9 since the insertion of a single flux quantum in the 3/8 BS state produces $n_q = 6$ fundamental quasiholes each of charge $e/16$ [18].

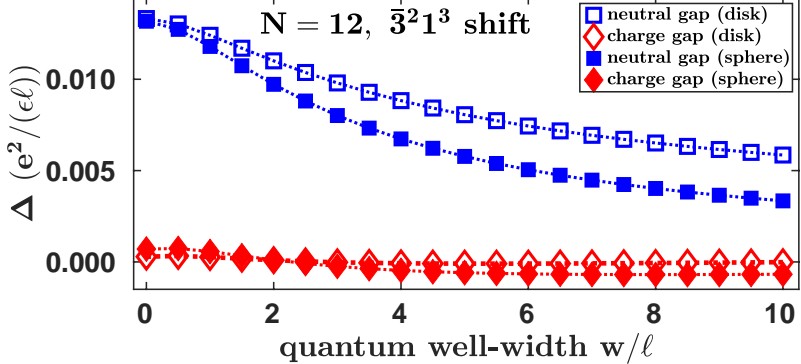

Figure 9: (color online) The second Landau level charge (red diamonds) and neutral (blue squares) gaps for $N = 12$ electrons at the shift corresponding to the $\bar{3}^2 1^3$ state at $\nu = 3/7$ evaluated in the spherical geometry using the spherical (filled symbols) and disk pseudopotentials (open symbols) for various well-widths $w$.

In summary, our results encouragingly suggest that the $\bar{3}^2 1^3$ state gives a good description of the $2 + 3/7$ Coulomb ground state observed in numerics. However, we find that the $\bar{3}^2 1^3$ state has a very small to vanishing charge gap. This suggests that the FQHE at $\nu = 2 + 3/7$ is very delicate state and provides a clue as to why it has not been definitively established in experiments.

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
