# Peer review of "A non-Abelian parton state for the $ν=2+3/8$ fractional quantum Hall effect"

_SciPost Physics, doi:SciPost Phys. 10, 083 (2021)_

## Round 1 · Referee Report · Gunnar Moller (Referee 1) · 2021-1-22

Strengths

1- comprehensive analysis of the CF parton wave function for the nu=2+3/8 quantum Hall effect, including quantum Monte-Carlo and exact diagonalization calculations
2- discussion of experimental signatures distinguishing the proposed trial state from alternative wave functions
3- nice review of the relevant background material
4- additional results on related wave functions presented in the appendix

Weaknesses

1- Some features of the proposed candidate wave function show unsatisfactory agreement with the exact diagonalization results: in particular:
a) the pair wave function has a pronounced shoulder at short distances, which is absent in the exact results
b) Its trial energy is higher than that of the competing Bonderson-Slingerland state

Report

The paper provides a comprehensive analysis of the different trial wave functions for the FQHE at nu=2+3/8, as well as discussing their topological properties. The results are a valuable addition to the relevant literature, and the paper is suitable for publication in SciPost.

While I agree with the author's argument that the trial wave functions should not be judged uniquely on the basis of accuracy of the energy, I wonder if the proposed state corresponds to a sufficiently robust quantum Hall state. Given the numerics on the sphere at N=12, and flux 2l=35, the correlation function of the ground state shown in Fig. 3 does reveal a large degree of oscillations even at large particle separation. This may well be a signature of instability. In order to reassure me in this respect, the author should add corresponding data for the aternative topological sectors, also, displaying the correlation functions for the shift of the competing Bonderson-Slingerland wave function.
The level of agreement between the correlation functions of the proposed parton state and the exact eigenstate should also be put in relation to that obtained for other trial wave functions in the second Landau level. For example consider the Moore-Read state where excellent agreement is reached at short distances, especially when the pair wave function is optimized (see Moller, G. & Simon, S. H. Paired composite-fermion wave functions. Phys. Rev. B 77, 075319 (2008)).

Some additional consideration should be given to these points:

- The author states that the overlap with the exact state or the entanglement spectrum could not be given, as they do not possess a Fock space representation. However, both these quantities can be efficiently evaluated using Monte-Carlo methods, also. The overlap calculation is standard, and for the ES, see e.g.: Rodriguez, I. D., Simon, S. H. & Slingerland, J. K. Evaluation of Ranks of Real Space and Particle Entanglement Spectra for Large Systems. Phys. Rev. Lett. 108, 256806 (2012).

-In the background section, it felt odd that no reference was made to the relevant trial states for the nu=3/8 state in the lowest Landau-level. Indeed, it appears to me that the 3/2 filled composite fermion LL was initially proposed as a candidate in the LLL.

Requested changes

1- Add a comparison on the correlation functions for the exact ground state at the shift of the Bonderson-Slingerland wave function.
2- Quantify arguments about geometries for N=12(18) being (in)accessible by quoting the relevant Hilbert space dimensions of the largest Lz subspace.
3- Add a calculation of the overlap for the parton state with the exact ground state, and possibly its entanglement spectrum
4- Amend Fig. 2 so that figure captions do not overlap with axis ticks to increase clarity.

  • validity: high
  • significance: good
  • originality: good
  • clarity: top
  • formatting: excellent
  • grammar: perfect

Author:  Ajit Coimbatore Balram  on 2021-01-23  [id 1176]

(in reply to Report 1 by Gunnar Moller on 2021-01-22)
Category:
answer to question
pointer to related literature

Thanks for carefully reading the manuscript, suggesting changes, and recommending publication. Here is a detailed response to the comments:

In the background section, it felt odd that no reference was made to the relevant trial states for the nu=3/8 state in the lowest Landau-level. Indeed, it appears to me that the 3/2 filled composite fermion LL was initially proposed as a candidate in the LLL. The reason for leaving out the 3/8 state in the LLL (which as the referee points out can be described as the 3/2 filled composite fermion LL state) is that the most recent experiments (see Samkharadze et. al. https://journals.aps.org/prb/abstract/10.1103/PhysRevB.91.081109 and Pan et al. https://journals.aps.org/prb/abstract/10.1103/PhysRevB.91.041301) suggest that there is no FQHE in the LLL at 3/8. Incompressibility between 1/3 and 2/5 in the LLL has only been established at filling factors 4/11 and 5/13.

Requested changes: 1- Add a comparison on the correlation functions for the exact ground state at the shift of the Bonderson-Slingerland wave function. This data is already available in the literature (see Fig. 1 of https://journals.aps.org/prb/abstract/10.1103/PhysRevB.95.125302) and has now been referred to in the text. The pair wave function of the exact SLL Coulomb ground state at the BS shift also shows oscillations at large distances (though less pronounced than that at the proposed parton shift [in terms of Hilbert space dimensions, the system at the BS shift is smaller than that at the parton shift]). Note that the BS pair wave function also has a pronounced shoulder at short distances, which is again absent in the exact results.

2 - Quantify arguments about geometries for N=12(18) being (in)accessible by quoting the relevant Hilbert space dimensions of the largest Lz subspace. We have now stated the relevant Hilbert space dimensions for the N=12 and 18 systems in the text.

3- Add a calculation of the overlap for the parton state with the exact ground state, and possibly its entanglement spectrum The Monte Carlo estimate of the overlap of the trial state with the exact SLL Coulomb ground state is 0.63(4). This estimate has now been added to the text. The reason for not including the real-space entanglement spectrum (ES) is that for states carrying modes in both the forward and backward directions that is not described as the ground-state of a model Hamiltonian (our parton state is of this kind), it is hard to glean much information from the ES (see, for example, https://journals.aps.org/prb/abstract/10.1103/PhysRevB.95.125302).

4- Amend Fig. 2 so that figure captions do not overlap with axis ticks to increase clarity. The figure captions do not overlap with the axis ticks. Does the referee mean axis labels instead of figure captions? We have provided a vector graphics version of Fig. 2 which can be zoomed-in for clarity.

Intended changes: 1) Include a Monte Carlo estimate of the overlap of the proposed parton state with the exact SLL Coulomb ground state. 2) Include Hilbert space dimensions of the N=12 and 18 systems. 3) Add the Monte Carlo estimate of the overlap of the parton state with the exact SLL Coulomb ground state. 4) Add a reference to G. Moller, S. H. Simon, Paired composite-fermion wave functions. Phys. Rev. B 77, 075319 (2008).

---

## Round 2 · Referee Report · Anonymous (Referee 2) · 2021-3-8

Strengths

  1. state-of-the-art exact diagonalization results for $\nu=2+3/8$ FQHE
  2. pedagogical presentation of the theory and experiments including extensive up-to-date bibliography
  3. suggestion on how to test the states in experiments is provided

Weaknesses

  1. no estimates and/or discussion for omission of the effects of Landau Level mixing and spin-polarisation
  2. strong finite-size effects as can be seen in the correlation function
  3. there is a rather small overlap between the wave-function obtained using ED and the suggested exact wave-function

Report

The author reports on the state-of-the-art results of numerical calculations including exact diagonalization and Monte Carlo for the $\nu=2+3/8$ FQHE state observed in experiments and suggests an exact wave-function for this state. Comparison of the wave-function with the ED results, and with the BS-state is presented. Additional extensive numerical results for other FQHE fractions are provided in Appendices.

The results presented in the paper should be useful to the quantum Hall community. However, it is hard to judge their relevance to experiments.
The states found in experiments are known to be fragile and one would expect that LL mixing and electron spin (etc.) should be relevant for the
discussion of these states, which is also supported by the fact that the calculated energies of different candidate states are very close, and the
finite-size gaps are small. Unfortunately the author only shows the dependence of the gaps on the quantum-well width, while the discussion of
the spin and the LL mixing is omitted. From the theoretical perspective there is also a few problems, in my opinion. For example, there are strong
finite-size effects in the ED calculations, which can be seen in the results for the correlation function at small and the large distances.
In addition, the overall shape of the correlation function is quite far from the one calculated for the proposed exact state. Second,
the exact wave-functions suggested by the author are not the ones used in the calculations. Although the author does explain that this
is due to numerical difficulties, it would be good to see some theoretical justification.

While the paper has some problems which stem from numerical difficulties , I believe that it is a nice work which should be published, and I recommend this paper for publication in SciPost.

Requested changes

  1. perhaps make it more precise in the abstract which states have been studied (Eq. (3) and (4)), and mention other assumptions made in the paper
  2. add discussion/estimates of the effects of LL mixing and inclusion of spin degrees of freedom
  3. present the value for the wave-function overlap between the states found in ED and the BS state, as well as the value of the overlap between BS and 321 state.
  4. there seem to be a typo in v2 of the paper. on page 6 change "The charge and neutral gaps for N=12" to "The neutral and charge gaps for N=12".
  5. provide some explanation for the behaviour of the correlation function in Fig. (3) at large distances, calculated using ED, that shows a large oscillation around r/l=12. may be compare this result with the result with the one for smaller number of particles, if possible.
  6. in Fig. (4) while the charge gap seem to almost vanish at large values of the quantum well widths, the wave-function overlap given by the author is around 95 percent. it would be good to have a better understanding why finite-width corrections do not alter significantly the wave-function overlap

  • validity: good
  • significance: good
  • originality: good
  • clarity: high
  • formatting: excellent
  • grammar: excellent

Author:  Ajit Coimbatore Balram  on 2021-03-09  [id 1292]

(in reply to Report 2 on 2021-03-08)

Thanks for carefully reading the manuscript and recommending its publication.
Below is a detailed response to the issues raised. R is for the referee and A is for the author.

R: There is a rather small overlap between the wave-function obtained using ED and the suggested exact wave-function.
A: The overlap of the suggested wave function with the ED state is comparable to that of other candidate states in the second Landau level. For comparison, the 12-particle Laughlin state has an overlap of 0.5 with the 7/3 Coulomb state.

R: Second, the exact wave-functions suggested by the author are not the ones used in the calculations. Although the author does explain that this is due to numerical difficulties, it would be good to see some theoretical justification.
A: This is an excellent question, to which as far as I know, there is no rigorous justification. Nevertheless, as pointed out in the text (and in the work of Ref. [39]), there is strong numerical evidence to suggest that the form of the wave function used in out calculations lie in the same phase as the exact wave function. Also, a priori, there is no particular motivation or reason to choose one wave function over the other. Thus, I use the form that is amenable to a numerical evaluation which then allows for a comparison with the ED results.

Requested changes
R: perhaps make it more precise in the abstract which states have been studied (Eq. (3) and (4)), and mention other assumptions made in the paper.
A: SciPost Physics has a strict limit of 8 lines with regards to the length of the abstract and this article’s abstract is just within that limit. SciPost also recommends ``emphasizing the context, stating the problem studied, the methods used, the results obtained, the conclusions reached, and the outlook” in the abstract. Thus, we have focused mainly on the new parton state in the abstract and given other details in the main text.

R: add discussion/estimates of the effects of LL mixing and inclusion of spin degrees of freedom
A: Thanks for this suggestion. LL mixing can be studied, but is a major project in itself and would thus be worth looking into in the future. A discussion of spinful or two-component states has been given in the paragraph just before the section on Conclusions. This discussion has been kept brief and is at a qualitative level since to the best of my knowledge there is no conclusive evidence for the existence of non-fully polarized ground states in the second Landau level.

R: present the value for the wave-function overlap between the states found in ED and the BS state, as well as the value of the overlap between BS and 321 state.
A: The overlap between the ED state and the BS state was already given in Ref. 18 and has now been included in the text. The BS state has an overlap of 0.8 with the exact SLL Coulomb state, which is higher than the overlap of the parton state with the exact SLL Coulomb ground state (this is also anticipated from the fact that the BS state has lower energy than our parton state in the SLL). Since the BS state and the \bar{3}\bar{2}^{2}1^{3} state occur at different shifts, their overlap is zero.

R: there seem to be a typo in v2 of the paper. on page 6 change "The charge and neutral gaps for N=12" to "The neutral and charge gaps for N=12".
A: Indeed, this was a typo. Many thanks for pointing it out. It has now been fixed in the updated version of the manuscript.

R: provide some explanation for the behaviour of the correlation function in Fig. (3) at large distances, calculated using ED, that shows a large oscillation around r/l=12. may be compare this result with the result with the one for smaller number of particles, if possible.
A: This behavior can be attributed to finite-size effects and the curvature effects in the spherical geometry. Such oscillations have also been seen at other SLL fillings (see Ref. [23]). Unfortunately, the smallest and the only system accessible to ED for which the state of interest can be compared against is the system of 12 electrons.

R: In Fig. (4) while the charge gap seem to almost vanish at large values of the quantum well widths, the wave-function overlap given by the author is around 95 percent. it would be good to have a better understanding why finite-width corrections do not alter significantly the wave-function overlap.
A: The charge gaps are quite low and stay close to zero for all widths. The charge gaps are affected more strongly by finite-size issues as compared to the ground-state overlaps (or even neutral gaps) since the charge gaps are obtained from results of three systems, N@2Q-1 [quasiparticle], N@2Q [ground-state], and N@2Q+1 [quasihole], where N is the number of electrons and 2Q is the number of flux quanta (at which the ground-state occurs), while the ground-state (and also the neutral gap) calculation involves only a single system.

Intended changes:
Include the overlap of the 3/8 Bonderson-Slingerland state with the second Landau level Coulomb ground state.
Fix the typos and update references.

---

## Round 2 · Author Response

This is a revised manuscript in response to Referee 1's comments.

---

## Round 2 · List of Changes

1) Included a Monte Carlo estimate of the overlap of the proposed parton state with the exact SLL Coulomb ground state. 2) Included Hilbert space dimensions of the N=12 and 18 systems. 3) Added the Monte Carlo estimate of the overlap of the parton state with the exact SLL Coulomb ground state. 4) Added a reference to G. Moller, S. H. Simon, Paired composite-fermion wave functions. Phys. Rev. B 77, 075319 (2008). 5) Added additional results on related wave functions in an appendix.

---

## Round 3 · List of Changes

Included the overlap of the 3/8 Bonderson-Slingerland state with the second Landau level Coulomb ground state.
Fixed typos and updated references.
Fixed typos and updated references.

---

## Editorial Decision

published